# Impact of Variants in the *ATIC* and *ARID5B* Genes on Therapeutic Failure with Imatinib in Patients with Chronic Myeloid Leukemia

**DOI:** 10.3390/genes13020330

**Published:** 2022-02-10

**Authors:** Karla Beatriz Cardias Cereja Pantoja, Tereza Cristina de Brito Azevedo, Darlen Cardoso de Carvalho, Natasha Monte, Amanda de Nazaré Cohen Paes, Maria Clara da Costa Barros, Lui Wallacy Morikawa Souza Vinagre, Ana Rosa Sales de Freitas, Rommel Mario Rodríguez Burbano, Paulo Pimentel de Assumpção, Sidney Emanuel Batista dos Santos, Marianne Rodrigues Fernandes, Ney Pereira Carneiro dos Santos

**Affiliations:** 1Núcleo de Pesquisas em Oncologia, Universidade Federal do Pará, Belém 66073-005, PA, Brazil; karlacereja.ufpa@gmail.com (K.B.C.C.P.); darlen.c.carvalho@gmail.com (D.C.d.C.); ntshmonte@gmail.com (N.M.); acohencastro@gmail.com (A.d.N.C.P.); luivinagre@gmail.com (L.W.M.S.V.); anarosasff@gmail.com (A.R.S.d.F.); assumpcaopp@gmail.com (P.P.d.A.); npcsantos.ufpa@gmail.com (N.P.C.d.S.); 2Hospital Ophir Loyola, Belém 66063-240, PA, Brazil; terbazevedo@yahoo.com.br (T.C.d.B.A.); rommel@ufpa.br (R.M.R.B.); 3Laboratório de Genética Humana e Médica, Instituto de Ciências Biológicas, Universidade Federal do Pará, Belém 66077-830, PA, Brazil; mariacbarros99@gmail.com (M.C.d.C.B.); sidneysantosufpa@gmail.com (S.E.B.d.S.)

**Keywords:** chronic myeloid leukemia, imatinib, *ATIC* gene, *ARIDB5* gene, pharmacogenomics

## Abstract

Chronic myeloid leukemia (CML) is a myeloproliferative neoplasm derived from the balanced reciprocal translocation of chromosomes 9 and 22 t (9q34 and 22q11), which leads to the formation of the Philadelphia chromosome and fusion of the *BCR-ABL* genes. The first-line treatment for CML is imatinib, a tyrosine kinase inhibitor that acts on the BCR-ABL protein. However, even though it is a target-specific drug, about 25% of patients do not respond to this treatment. The resistance mechanisms involved in this process have been investigated and studies have shown that germinal alterations can influence this mechanism. The aim of this work was to investigate 32 polymorphisms in 24 genes of carcinogenic pathway to verify the influence of these genetic variants on the response to treatment with imatinib. Our results demonstrated that individuals with the recessive GG genotype for the rs2372536 variant in the *ATIC* gene are approximately three times more likely to experience treatment failure with imatinib (*p* = 0.045, HR = 2.726, 95% CI = 0.9986–7.441), as well as individuals with the TT genotype for the rs10821936 variant in the *ARID5B* gene, who also have a higher risk for treatment failure with imatinib over time (*p* = 0.02, HR = 0.4053, IC 95% = 0.1802–0.911). In conclusion, we show that variants in the *ATIC* and *ARIDB5* gene, never screened in previous studies, could potentially influence the therapeutic response to imatinib in patients treated for CML.

## 1. Introduction

Chronic myeloid leukemia (CML) has as its main characteristic the reciprocal and balanced translocation of chromosomes 9 and 22 t (9q34 and 22q11), which results in the Philadelphia chromosome and fusion of the *BCR-ABL* genes [1,2]. The expression of this gene is constitutive with tyrosine kinase activity, and it is responsible for leukemogenesis and maintenance of carcinogenic activity in CML [3,4].

Treatment of CML is performed with the tyrosine kinase inhibitor (TKI), imatinib, which acts directly on the BCR-ABL protein by decreasing its intracellular activity and, thus, controlling the carcinogenic environment [5]. However, even though this is a target-specific drug, the response to this treatment is variable, therefore, about 25% of patients are not responsive to imatinib [4,6].

Variations in response to treatment can be influenced by several factors, including alterations in drug metabolism genes. Genetic variants can alter gene expression and, thus, modulate the interaction of the expressed protein with the drug, making the response inefficient [7,8,9,10].

Another factor that can interfere with treatment responses is the genetic composition of a population. It is known that the responses among populations vary worldwide, due to different frequencies of genetic variants in genes involved in absorption, distribution, metabolism, and excretion (ADME) of drugs [11,12,13]. In addition, in this study, the population investigated is highly mixed, and it is important to carry out a genomic control based on genetic ancestry, therefore, that there is no population substructuring and ancestry is not a confounding factor in the analyses [14,15].

Thus, this study investigated 32 polymorphisms in 24 carcinogenic pathway genes: *ABCC1*, *ABCC2*, *ABCC3*, *AMPD1*, *ARID5B*, *ATIC*, *CCND1*, *CDKN2A*, *CEBPE*, *GGH*, *IKZF1*, *ITPA*, *MTHFD1*, *MTHFR*, *MTRR*, *NALCN*, *NOS3*, *PIP4K2A*, *SHMT1*, *SLCO1B1*, *SLCO1B3*, *TLR4*, *TNFAIP3*, and *TPMT*, aiming to verify the influence of these genetic variants on the response to treatment with imatinib.

## 2. Materials and Methods

### 2.1. Ethics, Consent, and Permissions

This study was approved by the Research Ethics Committee of the participating institutions, at the Ophir Loyola Hospital under license number 1.575.920/2016 and at the Núcleo de Pesquisas em Oncologia (NPO) under protocol number 3.354.571/2019. All participants agreed to participate in the research and signed an informed consent form allowing the use of their clinical and genetic data.

### 2.2. Investigated Population

We investigated a total of 165 patients diagnosed with CML, followed for at least 1 year of treatment at Hospital Ophir Loyola, a reference hospital in the onco-hematology service in the city of Belém do Pará, in Northern Brazil. All patients started treatment with imatinib mesylate and had a detailed clinical follow-up. For the analyses, the patients were divided into two groups: patients who responded well to treatment and patients who did not respond well to treatment. The criteria used to define the hematologic and molecular response followed the National Comprehensive Cancer Network [16].

### 2.3. Selected Markers

For marker selection, the criteria were based on PharmGKB, NCBI, and Ensembl databases, as well as data available in literature regarding important variables for the carcinogenic pathway. The description of markers can be found in the Appendix A.

### 2.4. DNA Extraction and Quantification

Genetic material was extracted from peripheral blood collected in EDTA tubes and using an Axy PrepTM Blood Genomic DNA Miniprep kit (Axygen Biotechnology, San Francisco, CA, USA), following the manufacturer’s instructions. The DNA concentration and purity were measured using a NanoDrop 1000 spectrophotometer (Thermo Scientific NanoDrop 1000, NanoDrop Technologies, Wilmington, DE, USA).

### 2.5. Genotyping

Genotyping of the samples was performed on a QuantStudio™ 12K Flex Real-Time PCR system (Applied Biosystems, Life Technologies, Carlsbad, CA, USA) using real-time PCR technology (TaqMan OpenArray Genotyping) by allelic discrimination, following all the manufacturer’s recommendations. 

### 2.6. Genetic Ancestry

An ancestry analysis was performed as described by Ramos et al. [17] using 61 autosomal ancestry informative markers (AIMs) in three multiplex PCR reactions, aiming to accurately estimate the individual and global interethnic mix [14]. Amplicons were analyzed using an ABI Prism 3130 sequencer (Thermo Fisher Scientific, Waltham, MA, USA) and Gene Mapper ID v.3.2 software (Thermo Fisher Scientific, Waltham, MA, USA). The proportions of individual genetic ancestors were estimated using the STRUCTURE v.2.3.3 software (Stanford University, Stanford, CA, USA), assuming three parental populations. This analysis was performed to control a possible population substructure, as the investigated population was highly mixed.

### 2.7. Statistical Analysis

To be included in the statistical analyses, the genotyping data needed at least 70% in coverage. The allelic and genotypic distribution is shown in Appendix A.

The statistical analyses were run in SNPassoc library in RStudio v.3.6.1 software (Boston, MA, USA). Differences in the categorical variable (sex) were tested using Pearson’s chi square, while the quantitative variable (mean age) was evaluated using Student’s *t*-test. The ancestry indices were compared between the groups using the Mann–Whitney test. Multiple logistic regressions were used to assess possible associations between the polymorphisms and the response to treatment with imatinib, by estimating the odds ratios (ORs) and their 95% confidence intervals (CIs). The Kaplan–Meier survival analysis was used to estimate possible differences in the time of loss of response for each genotype by estimating the hazard ratio (HR). For this statistical test, we evaluated the variable “time of treatment failure” (TTF)—an event in which treatment was changed, due to absence of molecular or cytogenetic responses or intolerance to treatment. A significance level of *p* < 0.05 was considered for all the statistical analyses.

## 3. Results

The clinical epidemiological data are shown in Table 1, showing that, among the 165 patients included in this study, 103 (63.1%) patients had an excellent response to treatment and 62 (36.9%) patients did not. When comparing the variable “age at diagnosis” between the groups (responders and non-responders) we found no significant difference (*p*-value = 0.451), the same result was found when gender distribution was analyzed (*p*-value = 0.078). Regarding the genetic ancestry in both groups, we also found no significant difference between them.

Further information on clinical data reveals that 34 (20.2%) patients failed to respond during treatment (mean 47.11 months) and 12 (7.14%) patients were unresponsive from the beginning of treatment. 

### 3.1. Genotype and Imatinib Response-Relative Risk Assessment (OR)

Regarding genetic variants, we found no association between responsiveness to therapy at a specific time and the investigated SNPs (Appendix A).

### 3.2. Time of Treatment Failure (TTF)/Risk Analysis over the Response Time (HR)

We investigated the relationship between genetic variants and the variable “time of treatment failure” (TTF) to estimate the risk of treatment failure over time. Our results indicate that the SNVs rs2372536 in the *ATIC* gene and the rs10821936 in the *ARID5B* gene were statistically significant (Table 2). 

According to our results, we can infer that individuals with the recessive GG genotype for the rs2372536 variant in the *ATIC* gene are approximately three times more likely to experience treatment failure with imatinib (*p* = 0.045, HR = 2.726, 95% CI 0.9986–7.441), as well as individuals with the TT genotype for the rs10821936 variant in the *ARID5B* gene, who also have a higher risk for treatment failure with imatinib over time (*p* = 0.02, HR = 0.4053, IC 95% 0.1802–0.911) (Figure 1).

## 4. Discussion

Imatinib (STI-571is a 2-phenylamino-pyrimidine compound that inhibits the autophosphorylation of the BCR-ABL protein. This happens through the binding of TKI to the ATP receptor (adenosine triphosphate), which does not allow the binding of the phosphate group of the ATP molecule, keeping the protein inactivated. In this way, the entire downstream signaling cascade is turned off and the leukemic cells stop dividing [4,5].

This drug is used primarily for the treatment of CML, gastrointestinal stromal tumors (GISTs), and Philadelphia chromosome-positive acute lymphoblastic leukemia (ALL Ph+) [18]. Imatinib was the first TKI that showed efficiency in the treatment of CML, and it was approved by the Food and Drug Administration (FDA) of United States of America and also by Agência Nacional de Vigilância Sanitária (Anvisa of Brazil) in 2001, soon beginning to be used as a first-line treatment for CML.

In addition to inhibiting the BCR-ABL protein, imatinib also works as an inhibitor of other signaling pathways, such as those activated by the platelet-derived growth factor receptor (PDGFR), c-Kit (type III member of kinase receptors), MAPK (mitogen-activated protein kinase), and PI3K/AKT (phosphatidyl inositol 3 kinase), thus, acting in several ways to block cell division [19].

Resistance to imatinib therapy occurs in about 25% of patients with chronic myeloid leukemia; the mechanisms involved in this process have been investigated and studies show that genetic and epigenetic alterations can influence it [7].

In this study, genetic variants in genes involved in the carcinogenic pathway drugs were investigated in order to understand their influence on the response to treatment with imatinib in patients with CML. Our results demonstrated a significant association between the rs2372536 of the *ATIC* gene and rs10821936 of *ARID5B* gene treatment failure with imatinib.

### 4.1. ATIC

The *ATIC* gene (5-aminoimidazole-4-carboxamide ribonucleotide formyltransferase/IMP cyclohydrolase) it is located on chromosome 2q35 and encodes a bifunctional enzyme that catalyzes the last two steps of purine biosynthesis, generating inosine monophosphate from the aminoimidazole carboxamide ribonucleotide. Its pathways are related to AMP-activated protein kinase signaling (AMPK) and to antifolate resistance, as it is able to convert (5-amino-1-(5-phospho-β-D-ribosyl) imidazole-4-carboxamide)) into 6-mercaptopurine ribonucleotide, an inhibitor of purine biosynthesis used in the treatment of leukemias [20].

The de novo purine synthesis pathway is important for the disordered tumor growth process, as it is part of an anabolic pathway of cell multiplication; it is widely used in metabolic reprogramming for cell survival [21,22]. However, the specific role of the *ATIC* gene in modulating cancer progression remains unknown [23,24,25].

This gene has been shown to be overexpressed in hepatocellular carcinoma (HCC) and related to a worse prognosis in this neoplasm. The authors found that ATIC activates mTOR-S6 kinase 1 signaling and consequently stimulates the proliferation and migration of oncotic cells [25]. Furthermore, the gene has been associated with autophagy and an increased risk of developing HCC [26], lung cancer [27], and multiple myeloma [28].

In addition, the *ATIC* gene is associated with the risk of lymphoma progression in cases of ATIC protein fusion with the protein of oncogene *ALK* (anaplastic lymphoma kinase) [29,30]. This fusion even influences the treatment because when ALK phosphorylates ATIC in Y104, there is an increase in enzymatic activity. ALK-mediated phosphorylation of ATIC can rescue cancer cells from cell death induced by antifolate agents [31]. These results together suggest that *ATIC* may play an important role in carcinogenesis and cancer cell survival even under treatment [32].

This type of relationship of *ATIC* gene variants has also been demonstrated in other investigations with cancer therapeutic resistance such as in the treatment of breast cancer with tamoxifen [33], the use of pemetrexed for non-small cell lung cancer [34], as well as the use of methotrexate for rheumatoid arthritis [35], pediatric osteosarcoma [36], and acute lymphoblastic leukemia [37]. The investigated rs2372536 polymorphism is a missense mutation, responsible for the substitution of a threonine for a serine at position 116 of exon 5 of the expressed protein (c.347C > G; Thr116Ser), and it is one of the main biomarkers investigated in the response to methotrexate in rheumatoid arthritis [38,39,40].

### 4.2. ARID5B

The *ARID5B* gene is part of the AT-rich interaction domain (ARID) family of DNA-binding proteins, which are described as chromatin remodeling factors and also responsible for regulating the transcription of target genes [41,42]. ARID5B forms a complex with the PHF2 protein, which has H3K9me2 histone demethylase activity. H3K9me2 is one of the main markers of silenced chromatin and, thus, there is an epigenetic regulation of gene expression [43].

In addition, recent findings demonstrate that ARID5B is involved in cell proliferation and acts in the growth and differentiation of progenitor B-lymphocytes. It is a co-activator that binds to the 5′-AATA(CT)-3′ sequence [43]. The rs10821936 is a variant located in intron 3, and variants present in this intron are the most associated with susceptibility to ALL [44,45,46,47]. Although its role in leukemogenesis is not fully understood, SNPs in intron 3 of *ARID5B* may alter the transcription network involving normal hematopoiesis, thus, altering cell growth and differentiation [48].

Variants in the *ARID5B* gene have also been related to ALL regarding relapse and treatment response [49]. It has also been associated with risk of developing colorectal cancer [50], participation in breast cancer metabolism [51], and with a protein with unregulated function in prostate cancer [52]. These findings suggest that this gene plays an important role in the carcinogenic process.

Therefore, we suggest that variants of the *ATIC* and *ARID5B* genes may interfere with imatinib response in patients with CML, once, even though these genes do not interact directly with the drug, they act in the cellular environment supporting the survival of cancer cells, thus, impairing the effect of the treatment.

## 5. Conclusions

We conclude that the never-before-screened genetic variants of the genes *ATIC* (rs2372536) and *ARID5B* (rs10821936) play a role in therapeutic failure with imatinib, the gold standard treatment for CML.

## Figures and Tables

**Figure 1 genes-13-00330-f001:**
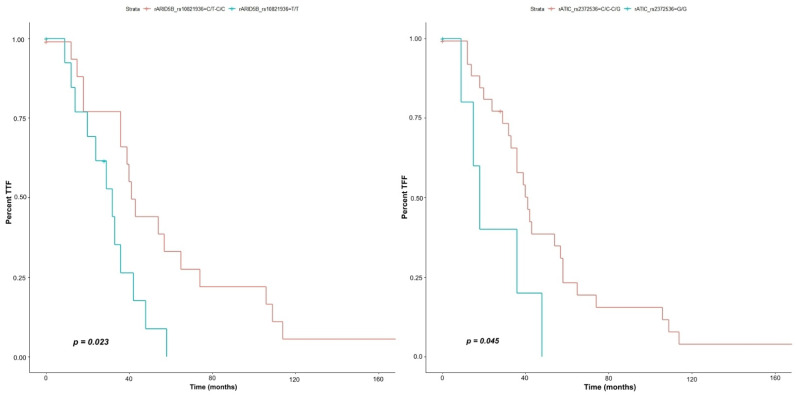
Kaplan–Meier curve demonstrating time of treatment failure (TTF) associated with rs10821936 of *ARID5B* gene and rs2372536 variant of *ATIC*.

**Table 1 genes-13-00330-t001:** Clinical and epidemiological variables of the investigated patients.

Variable	Responders	No Responders	*p*-Value
	103 (63.1%)	62 (36.9%)	
**Age (years)**	48.50 ± 15.07	46.76 ± 15.00	0.451
**Sex (%)**			0.078
Men	64 (62.1)	29 (46.7)	
Women	39 (37.9)	33 (53.3)	
**Ancestry Mean**			
European	0.468 ± 0.147	0.458 ± 0.147	0.797
Amerindian	0.299 ± 0.132	0.319 ± 0.133	0.604
African	0.233 ± 0.101	0.223 ± 0.101	0.686

**Table 2 genes-13-00330-t002:** Hazard ratio analysis of the genotypes analyzed with the time of treatment failure.

Genotype	HR (95% CI)	Lower	Upper	*p*-Value
*ATIC* rs2372536				
CC/CG vs. GG ^1^	2.726	0.9986	7.441	0.04
CC vs. CG + GG ^2^	1.484	0.7235	3.046	0.3
*ARID5B* rs10821936				
TT vs. CT + CC ^1^	0.4053	0.1802	0.911	0.02
CC vs. TT + CT ^2^	0.6114	0.2436	1.535	0.3
*TPMT* rs1142345				
TT + CT vs. CC ^1^	0.711	0.165	3.063	0.6
TT vs. CT + CC ^2^	0.9895	0.4341	2.255	1.00
*TPMT* rs12201199				
NA ^1,3^				
AA vs. AT + TT ^2^	0.6824	0.2561	1.818	0.4
*SLC01B1* rs4149056				
NA ^1,3^				
TT vs. CT + CC ^2^	1.109	0.3502	2.323	0.8
*ABCC2* rs717620				
NA ^1,3^				
CC vs. CT + TT ^2^	1.909	0.1807	1.518	0.2
*ABCC3* rs9895420				
NA ^1,3^				
TT vs. AT + AA ^2^	1.738	0.1983	1.67	0.3
*GGH* rs11545078				
NA ^1,3^				
GG vs. AG + AA ^2^	1.769	0.81	3.862	0.1
*GGH* rs3758149				
GG + AG vs. AA ^1^	0.9296	0.2162	3.997	0.9
GG vs. AG + AA ^2^	1.148	0.5384	2.446	0.7
*ATIC* rs4673993				
TT + CT vs. CC ^1^	2.561	0.3905	0.9388	0.06
TT vs. CT + CC ^2^	1.633	0.7888	3.382	0.2
*AMPD1* rs17602729				
NA ^1,3^				
GG vs. AG + AA ^2^	0.7357	0.2756	1.964	0.5
*CCND1* rs9344				
GG + AG vs. AA ^1^	1.536	0.5155	4.576	0.4
GG vs. AG + AA ^2^	1.174	0.5439	2.532	0.7
*IKZF1* rs4132601				
TT + GT vs. GG ^1^	0.7569	0.1013	5.657	0.8
TT vs. GT + GG ^2^	1.164	0.5258	2.579	0.7
*ITPA* rs1127354				
CC vs. AC ^1^	1.957	0.4465	8.575	0.4
*MTRR* rs1801394				
AA + AG vs. GG ^1^	0.8794	0.328	2.358	0.8
AA vs. AG + GG ^2^	0.9615	0.4536	2.038	0.9
*MTHFD1* rs2236225				
GG + AG vs. AA ^1^	0.6535	0.2462	1.735	0.4
GG vs. AG + AA ^2^	0.7404	0.3318	1.652	0.5
*NOS3* rs1799983				
NA ^1,3^				
GG vs. GT ^2^	1.052	0.2408	4.592	0.9
*MTHFR* rs1801133				
GG + GA vs. AA ^1^	1.445	0.4198	4.97	0.6
GG vs. GA + AA ^2^	0.9837	0.4521	2.141	1.00
*TLR4* rs4986790				
NA ^1,3^				
AA vs. AG ^2^	1.545	0.6471	0.2034	0.7
*TPMT* rs1800460				
CC vs. CT + TT ^1^	0.6339	0.218	1.843	0.4
NA^2^				
*SLCO1B1* rs4149015				
GG vs. AG + AA ^1^	0.8043	0.2271	2.849	0.7
AA vs. GG + AG ^2^	3.472	0.288	0.4051	0.2
*GGH* rs1800909				
AA+AG vs. GG ^1^	0.8698	0.2995	2.526	0.8
NA ^2^				
*NALCN* rs7992226				
AA+AG vs. GG ^1^	0.03471	0.42	2.552	0.9
AA vs. AG + GG ^2^	1.199	0.348	4.132	0.8
*SHMT1* rs1979277				
AA+AG vs. GG ^1^	0.873	0.3515	2.168	0.8
GG vs. AA + AG ^2^	3.361	0.6504	1.447	0.1
*SLCO1B1* rs2306283				
GG+AG vs. AA ^1^	1.733	0.6368	4.718	0.3
GG vs. AG + AA ^2^	1.276	0.7837	0.4663	
*CEBPE* rs2239633				
GG+AG vs. AA ^1^	1.899	0.8027	4.493	0.1
GG vs. AG + AA ^2^	2.229	0.4485	0.506	0.3
*TNFAIP3* rs6920220				
GG+AG vs. AA ^1^	0.6463	0.2426	1.722	0.4
NA ^2^				
*PIP4K2A* rs7088318				
AA vs. AC + CC ^1^	0.5205	1.921	0.1135	0.4
NA ^2^				

^1^ Recessive model; ^2^ Dominant model; ^3^ Not applicable for analysis.

## Data Availability

The data presently in this study are available on request from the corresponding author. The data are not publicly available due to the privacy topics contained in informed consent.

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
