# Peer review of "Impact of Variants in the ATIC and ARID5B Genes on Therapeutic Failure with Imatinib in Patients with Chronic Myeloid Leukemia"

_genes, 2022, doi:10.3390/genes13020330_

Round 1
Reviewer 1 Report
The manuscript by Cereja Pantoja and colleagues characterizes a link between genetic variants in drug metabolic genes and resistance to Imatinib, a front-line drug in the treatment of Chronic Myeloid Leukemia. They find that bi-allelic 347C-to-G SNP in ATIC associates with unfavorable response to Imatinib.
Manuscript has a clearly stated experimental goal. However, the manuscript lacks some important descriptions on the inclusion criteria for scrutinized gene panel. Also, it is not described how Authors, based on the previous research, would link the observed polymorphism to resistance to Imatinib.
Major concerns
- my major concern about the manuscript is lack of the discussion of what the observed SNP would mean to ATIC protein. What kind of the mutation it is (change of sense nonsense?) Which part of the gene is affected? Coding, non-coding, which domain of the protein? What is the predicted effect on the activity of ATIC (e. g. would it phenocopy gain of function mutations observed in hepatocellular carcinomas)? How Authors envision the action of the protein derived from the gene with SNP in terms of resistance to Imatinib? Are there any existing reports that may shed some light onto why bi-allelic G-G polymorphism of ATIC would result in the resistance to Imatinib in particular? These aspects should be included in the discussion.
- Inclusion criteria for the gene polymorphisms interrogated in the manuscript is insufficiently described. Have they been chosen based on the previously described action on drugs with some degree of structural homology with Imatinib? Is there any immediate justification for choosing this panel over any other xenobiotic metabolism-related genes?
- In my eyes the patients with adverse effects should not be included in the patient group employed for looking at the mechanisms of lack of response to the Imatinib. Mechanisms underlying the drug toxicity and resistance are likely different.
Minor comments
- Imatinib is an inhibitor of tyrosine kinase of BCR-ABL. To my knowledge, it does not reduce expression of the fusion protein as Authors state in line 43.
- Manuscript would greatly benefit from thorough re-reading (e. g. mistakes such as ‘e’ instead of ‘and’ in line 61, duplications of words such as ‘available’ in line 81, abbreviation ITQ in line 158 is not explained, etc.). Further from that, I would propose a proof-reading by a native English speaker.
Reviewer 2 Report
The manuscript entitled Impact of rs2372536 variant of the ATIC gene on therapeutic failure on the use of imatinib in patients with Chronic Myeloid Leukemia, written by Karla Beatriz Cardias Cereja Pantoja is well written and designed.
Minor comments
One graphical abstract may increased the interest of the readers,
The authors can perform additional analyses like one with all SNPs (regardless WHE results) and discuss them on discussion section.
Round 2
Reviewer 1 Report
Authors largely addressed my main concerns.